# Predictive Multi Experiment Approach for the Determination of Conjugated Phenolic Compounds in Vegetal Matrices by Means of LC-MS/MS

**DOI:** 10.3390/molecules27103089

**Published:** 2022-05-11

**Authors:** Eleonora Oliva, Federico Fanti, Sara Palmieri, Eduardo Viteritti, Fabiola Eugelio, Alessia Pepe, Dario Compagnone, Manuel Sergi

**Affiliations:** Faculty of Bioscience and Technology for Food, Agriculture and Environment, University of Teramo, Via Renato Balzarini 1, 64100 Teramo, Italy; eoliva@unite.it (E.O.); spalmieri@unite.it (S.P.); eviteritti@unite.it (E.V.); feugelio@unite.it (F.E.); apepe1@unite.it (A.P.); dcompagnone@unite.it (D.C.); msergi@unite.it (M.S.)

**Keywords:** polyphenols, HPLC-MS/MS, neutral loss scan, IDA, EPI

## Abstract

Polyphenols (PCs) are a numerous class of bioactive molecules and are known for their antioxidant activity. In this work, the potential of the quadrupole/linear ion trap hybrid mass spectrometer (LIT-QqQ) was exploited to develop a semi-untargeted method for the identification of polyphenols in different food matrices: green coffee, *Crocus sativus* L. (saffron) and *Humulus lupulus* L. (hop). Several conjugate forms of flavonoids and hydroxycinnamic acid were detected using neutral loss (NL) as a survey scan coupled with dependent scans with enhanced product ion (EPI) based on information-dependent acquisition (IDA) criteria. The presented approach is focused on a specific class of molecules and provides comprehensive information on the different conjugation models that are related to specific base molecules, thus allowing a quick and effective identification of all possible combinations, such as mono-, di-, or tri-glycosylation or another type of conjugation such as quinic acid esters.

## 1. Introduction

Polyphenols (PCs) are organic molecules present in plants and are known for their important antioxidant effects; they are mainly found in fruit, vegetables, cereals, olives, dried legumes, chocolate and some beverages, such as tea, coffee and wine [1].

PCs have been extensively studied for their potential in the prevention and treatment of diseases related to oxidative stress [2]. In fact, they can prevent the oxidative damage of biomolecules (DNA, lipids and proteins) induced by reactive oxygen species (ROS), such as superoxide anion, hydroxyl radical, nitric oxide, peroxides, etc.; the high production of oxidants species can cause various health problems [3], such as metabolic disorders, cancer, obesity, diabetes and cardiovascular disease [4].

PCs are generally grouped into two main categories based on their structure and properties: non-flavonoid compounds, such as phenolic acids, stilbenes and lignans, and flavonoids.

Phenolic acids can be further divided into two groups: benzoic acid derivatives called hydroxybenzoic acids (C_6_–C_1_), and cinnamic acid derivatives called hydroxycinnamic acids (C_6_–C_3_) [2]. Phenolic acids may also be found in conjugated forms; in fact, these acids can undergo various reactions, such as esterification, which leads to the formation of esters of hydroxyl acids or lipids, amidation, where the hydroxyl groups are conjugated with amino acids, peptides and mono or polyamines, or glycosylation [5]. In particular, the hydroxycinnamic acids are present, such as glucose esters or quinic acid esters.

Flavonoids (C_6_–C_3_–C_6_), as reported in reference [6], can be divided into six different main subclasses: flavanols (catechins), flavones, flavonols, flavanones, anthocyanins and isoflavones. Within each class, the B ring is generally linked to position 2 of the C ring, but it can also be linked to position 3 (in the case of isoflavones) or 4 (in the case of neoflavonoids); rings A and B can undergo different substitutions, such as oxygenation, alkylation, glycosylation, acylation and sulfonation, giving rise to the different conjugated compounds [7], and thus making flavonoids one of the largest and most diversified groups of phytochemicals in nature (Figure 1).

The most common sugar moieties of flavonoids present in plants are glucose, rhamnose, galactose, apiose, rutinoside, arabinose and glucuronic acid; in fact, flavonoids are usually available in the form of mono, di- and triglycosides [8]. As for monoglycosides, they are commonly known as *O*-glycosides and C-glycosides; in diglycosides, the sugar moieties are bound to the same or to two different carbon atoms. Triglycosides seem to have a lower occurrence [9].

It should be noted that the chemical structure of the flavonoids influences the biological activity, including antioxidant capacity [10]; thus, the characterization of the different subclasses of PCs is crucial to knowing the antioxidant potential of a plant and understanding the correlation between structure and activity.

For the importance of PCs, there are numerous analytical methods for their determination in food matrices; high-performance liquid chromatography (HPLC) coupled with mass spectrometry (MS) is certainly the most used method for their identification and quantification. MS was applied both in low resolution (LRMS) [11,12], using a triple quadrupole (QqQ), linear ion trap (LIT) and hybrid quadrupole-linear ion trap mass spectrometers (LIT-QqQ) [13] as the detectors, or in high resolution (HRMS) [14,15], coupled with instruments such as Q-TOF [16,17], Q-Orbitrap and LTQ-Orbitrap [18,19], which is usually used for comprehensive characterization by means of untargeted analysis.

The QqQ is usually limited by the necessity of reference standards in terms of compound identification; however, reliable qualitative information may be achieved using different data acquisition strategies.

By means of the information-dependent acquisition (IDA) approach, it is possible to design an experimental set-up allowing a semi-untargeted analysis that can provide structural information on specific classes of compounds even without analytical standards.

This approach involves the use of multiple scans with different acquisition modes in a single run; in particular, it is possible to work with one (or two) first scan (survey scan), which is used to analyze the sample on the basis of the information to be obtained. The IDA tool analyses the data during acquisition, depending on the pre-determined criteria, selecting the ions on which further experiments are run based on the previous scans (dependent scan).

Different scan modes can be used for the survey scan: multiple reaction monitoring (MRM), precursor ion scan (PIS), neutral loss (NL), enhanced mass spectrometry (EMS), enhanced multi-charged (EMC) or enhanced resolution (ER). They can be coupled with dependent MS/MS scans such as an enhanced product ion (EPI) to obtain a multi-component characterization.

For example, these acquisition methods have been mainly used for the determination of PCs by EMS-IDA-EPI [20,21,22,23], MRM-IDA-EPI [24,25] or for the determination of protonated and ammoniated mono-hexoside [26]. Furthermore, in HRMS, a similar acquisition modality is used, which exploits the full scan (FS) and the EPI for the determination of PCs [27,28,29].

In this work, targeted and semi-untargeted approaches are presented for identifying and quantifying different conjugated forms of PCs in different food matrices: green coffee, *Crocus sativus* L. (saffron) and *Humulus lupulus* L. (hop) were chosen to test the proposed strategy. Green coffee is a form of fruit of raw coffee, unroasted, unprocessed, and natural, known mainly for its high antioxidant properties due to the presence of numerous phenolic compounds, in particular phenolic acids and their derivatives, such as chlorogenic acids (CGA) [30,31]. Saffron, produced by the dried stigmas of *Crocus sativus* L., a member of the Iridaceae family, is a matrix rich in polyphenols [32]; in particular, the stigma is rich in numerous derivatives of flavonoids [33].

Hop is a plant belonging to the Cannabaceae family and contains luppolin glands, which produce groups of compounds that are precious for the brewing industry and secondary metabolites such as PCs. The interest in this matrix is mainly focused on the phenolic class of xanthones, xanthohumol and isoxanthohumol, characteristic of hop [34]. Due to the high content of polyphenols in these foods and the commercial importance, the proposed method was tested using these matrices.

The semi-untargeted method involves, for the first time, the use of a neutral loss (NL) IDA-enhanced production approach, implemented on an LIT-QqQ, and it is able to identify different glycosidic derivatives of flavonoids and quinic acid esters of hydroxycinnamic acids in the selected food matrices. NL scanning modes were used as survey scans for the identification of the common fractions of the different analytes. Then, using the IDA criteria, EPI-dependent experiments were carried out based on the data previously obtained in the NL scans.

## 2. Results

### 2.1. Targeted Analysis

MRM analysis was used for the identification and quantification of PCs in the three plant matrices, using standards with the previously described method [35]; the results are shown in Table 1. The analysis allowed us to identify and quantify both phenolic acids and flavonoids and their respective glycosidic derivatives. The method showed a sensitive and robust quantitative analysis of the target analytes, providing LOQs ranging between 0.0004 and 0.06 ng mg^−1^. Furthermore, the precision and accuracy of the method were suitable, with values included between ±10% near LOQs. The validation of the method was performed following the FDA guidelines [36], considering the LOQs, LODs, accuracy, precision and linearity parameters.

#### 2.1.1. Green Coffee Sample

The quantification of PCs in the green coffee sample showed a major portion of chlorogenic acid, corresponding to 93% of the total content of polyphenols [37]. Caffeic acid and ferulic acid make up 4% and 1.5%, respectively, of the total content [38]. Flavonoids such as epicatechin, catechin, rutin and quercetin have also been identified, present in a lower %, from 0.1% to 0.4%.

#### 2.1.2. Saffron Sample

The MRM analysis of the saffron sample showed high chlorogenic acid content, syringic acid and caffeic acid, which constitute 30%, 12% and 10% of the total PC content; other phenolic acids have also been identified, such as *p*-coumaric acid and gallic acid at 6% and 5%, respectively [39]. Among the flavonoids, quercetin is certainly the most relevant, with a total content of 20%, with myricetin and kaempferol both constituting 3%.

#### 2.1.3. Hop Sample

In the hop sample, the majority of the PC content is represented by flavonoids such as catechin and epicatechin, which constitute 31% and 5%, respectively, of the total content; from rutin, luteolin and quercetin, which correspond to 8%, 6% and 4% [40]. As far as phenolic acids are concerned, we find chlorogenic acid and protocatechuic acid, both of which constitute 6% of the total content of PCs, and *p*-coumaric acid and syringic acid, with 6% and 5%. Xanthones such as xanthohumol and isoxanthohumol are characteristic of this matrix and are present at 12% and 3% of the total content, respectively [41].

### 2.2. Semi-Untargeted Analysis

For the identification of the conjugated forms of the PCs, NL-IDA-EPI scans were then used with common losses both for the glycosidic forms of flavonoids and quinic acid esters of hydroxycinnamic acid, with neutral loss *m*/*z* of 132, 146, 162, 174, 176 and 308, corresponding to the mass of a pentose unit, rhamnose and coumaroyl units, a hexose and caffeoyl unit, quinoyl units, feruloyl unit, rutinose units, respectively; results are shown in Table 2, according to Clifford et al. [31] and Kolodziejczyk-Czepas et al. [42].

To assign the identity of the PCs found in the three food matrices, the fragmentation patterns obtained from the EPI scans were compared with the literature.

#### 2.2.1. Green Coffee

In green coffee samples, mainly quinic acid esters of hydroxycinnamic acids were identified; in particular, the NL scans revealed the presence of numerous quinic acid esters.

Indeed, as reported in Table 2, the NL-IDA-EPI strategy detected several neutral unit losses of 146 amu corresponding to the [M − H − coumaroyl]^−^ unit, 162 amu corresponding to the [M − H − caffeoyl]^−^ unit, 174 amu responding to thee [M − H − quinoyl]^−^ unit, and 176 amu responding to the [M − H − feruloyl]^−^ unit.

In fact, as reported by Alonso-Salces et al. [38], different isomers of quinic acid esters have been identified, such as isomers of caffeoylquinic acid (chlorogenic acid, cryptochlorogenic acid and neochlorogenic acid) and the isomers of feruloylquinic acid (Figure 2). Moreover, more complex forms, such as the isomeric forms of dicaffeoylquinic acid (cynarin and isochlorogenic acid) and isomers forms of diferuloylquinic acid [43], have been identified.

With NL scans of 162 amu, the catechin-hexoside has been identified, with the EPI experiment showing the fragmentation pattern *m*/*z* 109, 125, 245 and 289, predominant in the coffee varieties [44].

#### 2.2.2. Saffron

In saffron, more glycosylated, diglycosylated and triglycosylated forms were identified. In fact, using the neutral fragment 162 amu, numerous hexoside, glucosylated or galactosylated forms have been found.

In fact, combining the NL and EPI experiments allowed us to identify many conjugated forms of flavonoids, already known in saffron [45,46,47], such as dimethylquercetin, petunidin-hexoside, quercetin-hexoside, kaempferol-hexoside, kaempferol-sophoroside-hexoside, kaempferol-rutinoside-hexoside, quercetin-rutinoside, kaempferol-rutinoside, kaempferol-sophoroside, quercetin-sophoroside, isorhamnetin-sophoroside, as well as complex forms such as quercetin-*O*-(6″-acetyl-galactoside)-*O*-rhamnoside, isorhamnetin-rutinoside and kaempferol-(6″-acetyl-glucoside)-glucoside.

Moreover, other compounds have been identified with this approach but, to the authors’ knowledge, are not present in the literature.

With the neutral loss of the [M − H − rham]^−^ unit, kaempferol-rhamnoside was identified with a fragmentation pattern of *m*/*z* 133, 159, 229 and 285, and isorhamnetin-sophoroside-rhamnoside was identified, also with the [M − H − Gly]^−^ unit, with *m*/*z* 300, 315, 477, which corresponds to isorhamnetin-rhamnoside and 639, which is isorhamnetin-sophoroside.

Instead, with the common neutral fragment, 162 amu protocatechuic acid-hexoside (*m*/*z* 53, 109, 135, 153), cyanidin-hexoside (*m*/*z* 213, 231, 259 and 287) and pelargonidin-hexoside (*m*/*z* 141, 188, 225 and 270) were identified.

Several compounds have been identified with neutral losses of [M − H − 308]^−^, [M − H − 162]^−^ and [M − H − 146]^−^ as reported in Table 1; among these, the following compounds were detected in saffron for the first time: myricetin-rutinoside-hexoside with fragments *m*/*z* 179, 317, 463, which corresponds to myricetin-rhamnoside, 479 matches with the myricetin-hexoside, and 625 corresponds to myricetin-rutinoside.

Taking advantage of the NL detection of the apiosyl unit, [M − H − 132]^−^ and hexoside unit [M − H − 162]^−^, apigenin-apiosyl-hexoside was also identified (*m*/*z* 117, 151, 269 and 431) for the first time.

Finally, with the neutral losses, [M − H − 162]^−^ and [M − H − 308]^−^ kaempferol-glucosyl-(1->2)-(6″-acetylgalactoside)-hexoside (Figure 3) were identified; their EPI fragmentation shows *m*/*z* 489, which corresponds to kaempferol-(6″-acetylgalactoside), m/z 609 matches with kaempferol-sophoroside, *m*/*z* 651 corresponds to kaempferol-(6″-acetylgalactoside)-hexoside. It was also possible to identify a trihexoside [M − H − 162 − 162 − 162]^−^ kaempferol-sophoroside-hexoside with a fragmentation pattern of *m*/*z* 255, 285, 446 and 609, which corresponds to kaempferol-sophoroside.

#### 2.2.3. Hop

The same approach was used for the hop samples, where the NL-IDA-EPI analysis highlighted the presence of the three isomers of caffeoylquinic acid and the three isomers of sinapoylquinic acid identified by the neutral loss of 162 amu, as reported by Tang [48]. Furthermore, quercetin-hexoside, quercetin-rutinoside and kaempferol-rutinoside were also identified as a result of the common loss of the [M − H − 162]^−^ unit and [M − H − 308]^−^ unit, according to Maietti et al. [49].

Moreover, the neutral loss of the [M − H − 132]^−^ unit allowed us to identify two further analytes not yet found in this matrix, as far as we know, isorhamnetin-xyloside (Figure 4) and quercetin-xyloside, where the fragmentation pattern provided by EPI showed fragments *m*/*z* 151, 271, 300, 315 and *m*/*z* 151, 179, 271, 301, respectively, and with the neutral loss of the [M − H − 162]^−^ unit, caffeoyltartaric acid.

## 3. Discussion

The PCs in the three food matrices were identified using two different approaches; the MRM acquisition mode to identify and quantify 35 polyphenols in a single run and the NL-IDA-EPI strategy to identify conjugated forms. The coupling of QqQ with LIT eliminates the strong dependence of the QqQ-MS on the use of reference standards for the identification of the analyte; in fact, the comparison of the fragmentation patterns of the different phenolic conjugates can be easily achieved with online databases. This allows to putatively identify phenolic derivatives, both glycosylated and quinic acid esters, determining the structures based on known fragmentation patterns.

### 3.1. HPLC-MS/MS Method Development

The targeted approach was used to identify and quantify different PCs in the food matrices under examination. As reported in the literature, numerous studies have been carried out on green coffee beans [50,51] in which particular attention is paid to phenol acids and their derivatives such as the class of chlorogenic acids [31]; in saffron, the content of flavonoids and, in particular, conjugated forms has been mainly studied [52,53], and in hop, xanthones, in particular xanthohumol and isoxanthohumol, are the characteristic analytes [41,54].

The proposed approach allowed, with a single analysis, to identify and quantitate numerous PCs belonging to the different classes. The information derived from the semi-untargeted analysis allowed us to obtain a more complete characterization of the matrices.

The potential of LIT was exploited for the semi-untargeted analysis of PCs; however, as far as we know, this approach was not used for the polyphenols analyses in the three proposed matrices. Some studies were conducted with EMS-IDA-EPI and MRM-IDA-EPI acquisition modes [20,21,22,23,24,25], and a targeted list was used for compound identification.

In our work, with the neutral loss scan, it was possible to identify numerous compounds characterized by the same moiety, but belonging to different classes, i.e., the neutral fragment 162 Da allowed the detection of compounds with a glucose moiety in the structure, such as the glycosidic derivatives of flavonoids, and the derivatives of caffeic acid.

The NL-IDA-EPI acquisition mode was used to identify the numerous conjugated forms of PCs that have common neutral losses, such as the sugar fraction [M-H-Gly]^−^ and the quinic acid esters of hydroxycinnamic acids [M − H − quinoyl]^−^. Using the IDA criteria, with dynamic subtraction of the background of the survey scan, it was possible to activate the EPI scan and confirm the different analytes in the same LC/MS run.

Two NL survey scans were used, with a scan range of *m*/*z* 300–1000, coupled with four EPI experiments in the same range. The instrumental parameters needed a fine tuning in order to obtain the aimed performances; for example, the IDA threshold was set at 5000 cps, above which the fragmentation patterns of each analyte were collected by EPI of the four most intense peaks; the LIT fill time was set to 25 ms, and the mass tolerance was set at 0.25 Da. For the acquisition of the collision-induced dissociation spectra (CID), nitrogen was used as collision gas and a CE of –55 eV with CES ± 10 eV. For the DP, EP and CXP, the same parameters of the MRM analysis were used: –100 eV, –9 eV and –10 eV, respectively.

The EPI mode is characterized by a higher sensitivity than the classic scanning of the product ion (PI) as the accumulation of ions occurs in the LIT, and being faster than the classic quadrupole, it also allows us to obtain a lower cycle time. It also provides more information on the analyte thanks to the Collision Energy Spread (CES), which allows us to work on three different CEs, thus obtaining MS/MS spectra rich in characteristic fragments.

EPI, unlike PI, also allows us to work both in targeted and semi-untargeted mode, as the obtained fragmentation pattern can be used to identify the analyte through the interpretation of the numerous product ions obtained by comparison with mass spectral libraries, to confirm the identity of the analyte and discriminate false positives, such as potential co-eluting compounds.

For the method set-up, a group of 15 analytes was used for the selection of moieties used for the NL experiment (training set); the following analytes were included in this list: gallic acid, chlorogenic acid, tyrosol, catechin, quercetin, naringenin, hesperidin (hesperetin-7-*O*-rutinoside), apigenin, diosmetin (luteolin-4-methyl ether), orientin (luteolin-8-glucoside), quercetin, isoquercetin (quercetin 3-*O*-glucoside), rutin (quercetin 3-*O*-rutinoside), oleuropein, and xanthohumol. In the NL survey scan, several common losses were selected following neutral losses of quinic acid esters of hydroxycinnamic acids and glycosidic derivatives of flavonoids. For the quinic acid esters, the following values were used: 146 amu for [M − H − coumaroyl]^−^ unit, 162 amu for [M − H − caffeoyl]^−^ unit, 174 amu for [M − H − quinoyl]^−^ unit, and 176 amu for [M − H − feruloyl]^−^ unit. For the identification of the glycosidic derivatives, the following were used: the 132 amu for the [M − H − pentose]^−^ unit, the 146 amu also for [M − H − rham]^−^ unit, 162 amu also for hexose [M − H − hexose]^−^ unit and 308 amu for [M − H − rut]^−^ unit.

It was also possible to identify diglycosides and triglycosides with the same scan, as the fragments in common also correspond to multiple neutral losses [M − H − Gly − Gly]^−^ or [M − H − Gly − Gly − Gly]^−^.

The EPI experiments allowed the fragmentation of the precursor ion selected in Q_1_ and the ability to obtain the characteristic fragmentation patterns for each analyte.

A different set of molecules (validation set) was used to confirm the suitability of the developed method for the identification of the different conjugated forms of PCs. In this case, the following set of analytes were selected: OH-tyrosol, protocatechuic acid, 3-OH-benzoic acid, (−)-epigallocatechin gallate (EGCG), epicatechin, caffeic acid, vanillic acid, (−)-epigallocatechin (EGC), siringic acid, *p*-coumaric acid, hyperoside (quercetin-3-d-galactoside), ferulic acid, rosmarinic acid, *o*-coumaric acid, sinapic acid, myricetin, luteolin, *trans*-cinnamic acid, isoxhanthoumol, and kaempferol. The NL-IDA-EPI method positively detected all the analytes of the validation set, which were included in a mixed solution (100 ng, 75 ng and 25 ng mg^−1^), proving the reliability of the proposed approach.

For the analysis of real samples, the results obtained were compared with the spectra obtained from the standard mix, when available, or with databases for the putative identification of the molecules.

### 3.2. Polyphenols Identification

The two proposed approaches were tested on food matrices to evaluate their effectiveness. In the green coffee sample, it was possible to identify mainly derivatives of hydroxycinnamic acids, which were identified as the isomers of caffeoylquinic acid (chlorogenic acid, cryptochlorogenic acid and neochlorogenic acid) and more complex forms such as 1,5-dicaffeoylquinic (cynarin) acid and 3,4-dicaffeoylquinic acid (isochlorogenic acid) as well as derivatives of feruloylquinic acid, as reported in the literature [31,55,56]. In the saffron sample, an increase in conjugated forms of flavonoids, such as kaempferol-hexoside, pelargonidin-hexoside, petunidin-hexoside, and more complex forms, such as isorhamnetin-sophoroside, as known in the literature, was noted [57,58,59]. Finally, the MRM analyses carried out on the hop sample allowed us to identify and quantify xanthohumol and isoxanthohumol [41], which are characteristic of this matrix, while the semi-untargeted showed the presence of phenolic acids, such as the three isomers of caffeoylquinic acid [60] and flavonoid derivatives, such as kaempferol-rutinoside [61].

It can, therefore, be said that the proposed methods are able to provide reliable results comparable with the information present in the literature, thus allowing us to carry out characterizations of the matrices of interest with and without the use of standards.

## 4. Materials and Methods

### 4.1. Chemicals

The matrices used for the analysis of PCs were purchased from local resellers: green coffee and saffron in the form of powder, while hops as inflorescence.

The standard PCs used in our research were: gallic acid, OH-tyrosol, protocatechuic acid, (−)-epigallocatechin (EGC), 3-OH-benzoic acid, tyrosol, chlorogenic acid (3-*O*-caffeoylquinic acid), epicatechin, caffeic acid, vanillic acid, catechin, (−)-epigallocatechin gallate (EGCG), siringic acid, orientin (luteolin-8-glucoside), rutin, *p*-coumaric acid, hyperoside (quercetin-3-d-galactoside), isoquercetin (quercetin-3-b-d-glucoside), ferulic acid, hesperidin, rosmarinic acid, oleuropein, *o*-coumaric acid, sinapic acid, myricetin, luteolin, quercetin, *trans*-cinnamic acid, naringenin, isoxhanthohumol, apigenin, diosmetin (luteolin-4-methyl ether), kaempferol, and xanthohumol, which were purchased from Sigma-Aldrich (St. Louis, MO, USA). The working standard mixtures were prepared by appropriate dilution in methanol (MeOH) and stored at −20 °C.

Ultrapure water (H_2_O), acetic acid, ethanol (EtOH), MeOH and acetonitrile (ACN) were UPLC-MS grade and were purchased from VWR (Radnor, PA, USA).

### 4.2. Sample Preparation (Extraction)

The extraction of PCs from the green coffee sample was performed according to Ramón et al. [62] with some modifications: 10 mg of the sample were weighed and extracted with 1 mL of EtOH/H_2_O (80:20 *v*/*v*) by ultrasonic water bath for 30 min at 25 °C. The PCs from the saffron sample were extracted according to Ferrara et al. [53]; 10 mg of the sample were weighed and extracted with 1 mL of EtOH/H_2_O (50:50 *v*/*v*) by ultrasonic water bath for 30 min at 25 °C. The procedure proposed by Quir et al. [41] was used to extract the PCs from the hop samples: 10 mg of dried hop were weighed and extracted with 1 mL of MeOH in an ultrasonic water bath for 30 min at 25 °C. All samples are centrifuged at 10,000 rpm for 10 min, and 10 µL of each were loaded onto the SPE for the subsequent clean-up procedure.

### 4.3. SPE Extraction

For the clean-up phase of the matrices, the SPE with Strata XL cartridge (330 mg, 1 mL) by Phenomenex (Torrance, CA, USA) was used, as described in a previous study [35]. Briefly, the cartridges were activated with 1 mL of MeOH and subsequently conditioned with 1 mL of a phosphate buffer mixture (50 mM) at pH 3: MeOH (90:10 *v*/*v*). Each extract was diluted in 1 mL of the conditioning solution and loaded onto the cartridge. A total of 1 mL of H_2_O at pH 3 was used to wash the cartridge to remove the polar interfering compounds. Finally, the analytes were eluted with 1 mL of MeOH; 6 µL were injected into the HPLC-MS/MS system.

### 4.4. HPLC–MS/MS Analysis

The HPLC–MS/MS analysis of PCs was performed following Oliva et al. [35]. Briefly, a Nexera XR LC system (Shimadzu, Tokyo, Japan) was coupled to a Qtrap 4500 mass spectrometer (Sciex, Toronto, ON, Canada) equipped with a heated ESI source. Analytes were separated using an Excel 2 C18-PFP (10 cm × 2.1 mm ID) column from Advanced Chromatography Technologies (Aberdeen, UK) with 2 μm particles and a safety guard. H_2_O with 0.5% acetic acid (A) and can (B) were used as mobile phases. The injection volume and the flow rate were set at 6 µL and 0.300 mL min^−1^, respectively. All analytes were detected in negative ionization mode with a capillary voltage of −4500 V, nebulizer gas (air) at 40 psi and turbo gas (nitrogen) at 40 psi and 500 °C. With the MRM mode, the identification and quantification of ionic currents was performed for each analyte. The data collection and processing were performed with the Analyst 1.7.2 software and the quantification with the Multiquant 3.0.3 software (Sciex).

Conjugated forms of PCs, not present in the target list, were identified using the NL-IDA-EPI mode to elucidate their structures based on known fragmentation patterns. With the survey scan (NL), it was possible to select all precursor ions characterized by a common moiety, such as the glycosidic fraction in flavonoids or quinic acid esters moiety in hydroxycinnamic acids; in particular, the following ions were selected: 132 amu corresponding to the mass of a pentose, 146 amu for rhamnose and coumaroyl units, 162 amu a hexose, and caffeoyl groups, 174 amu for quinoyl groups, 176 amu for feruloyl unit, and 308 amu for rutinose unit.

The data obtained in the first phase of acquisition of the masses selected in the NL scan were then used to trigger EPI-dependent scans, using the IDA criteria pre-set, which made it possible to obtain characteristic fragmentation patterns for each analyte identified.

The MS^2^ conditions for the glycosylated forms included in the target list were used for ionization, transmission, and fragmentation parameters: DP and EP were set to −100 eV and −9 eV, respectively, while CE was set to −50 eV and CXP to −10 eV.

The conditions for the EPI experiment were evaluated based on the results obtained with the same principle of NL: DP—100 eV, EP—9 eV, CE—55 eV, with the addition of CES—10 eV.

## 5. Conclusions

Due to the structural characteristics of PCs, together with the esterification and amidation patterns of the hydroxyl groups of phenolic acids (in particular, the esters of quinic acid) and those of glycosylation and hydroxylation of the three flavonoid rings, this class of phenolic compounds represents one of the largest and most diversified groups of phytochemicals in nature, with a strict relation between the chemical structure and biological activity. For this reason, it is very important to elucidate the chemical structure and especially the conjugation pattern.

Different techniques are commonly used for the analysis of polyphenols, but LC-MS provides the best performance in terms of identification power based on the physico-chemical characteristics of this class of molecules. HRMS is currently the gold standard for its high capability of identifying a large number of compounds, but it often needs strong data mining software due to the huge data collected for the untargeted acquisition mode.

The first approach presented, ‘targeted’, allowed the identification and quantification of numerous PCs, while the second, ‘semi-untargeted’, is focused on a selected class of molecules, and it may provide exhaustive information on the different conjugation patterns that are related to a specific base molecule, thus allowing a rapid and effective identification of all the possible combinations, such as mono-di-tri-glycosylation or other kinds of conjugation (such as quinic acid esters, glycosides such as cyanidin-hexoside, xylosides such as isorhamnetin-xyloside, rhamnosides such as kaempferol-rhamnoside, rutinosides such as quercetin-rutinoside or sophorosides such as isorhamnetin-sophoroside-rhamnoside).

By means of a rational exploitation of LIT-QqQ, it was possible to obtain a putative identification of different conjugated forms of the main flavonoids; EPI and NL scan modes were combined by using an IDA approach in order to investigate the polyphenolic content of different vegetal matrices (green coffee, saffron, hop) and identify a large number of mono-, di- and tri-glycosylated forms, as well as esters of quinic acid. In particular, several compounds not belonging to the target list were detected in the selected matrices: isomers of caffeoylquinic acid (chlorogenic acid, cryptochlorogenic acid and neochlorogenic acid) and the isomers of feruloylquinic acid have been identified in green coffee, in addition to their complex forms (dicaffeoylquinic acid and diferuloylquinic acid); conjugates of flavonoids such as apigenin-apiosyl-hexoside and kaempferol-glucosyl-(1-> 2)-(6″-acetylgalactoside)-hexoside have been identified in saffron, while both conjugated forms of the quinic acid esters such as the isomers of sinapoylquinic acid and glycosylated flavonoids, e.g., isorhamnetin-xyloside, have been identified in hop.

In our opinion, the proposed approaches could represent a useful tool for the investigation of the phenolic compounds in food matrices with a specific focus on the different conjugated forms.

## Figures and Tables

**Figure 1 molecules-27-03089-f001:**
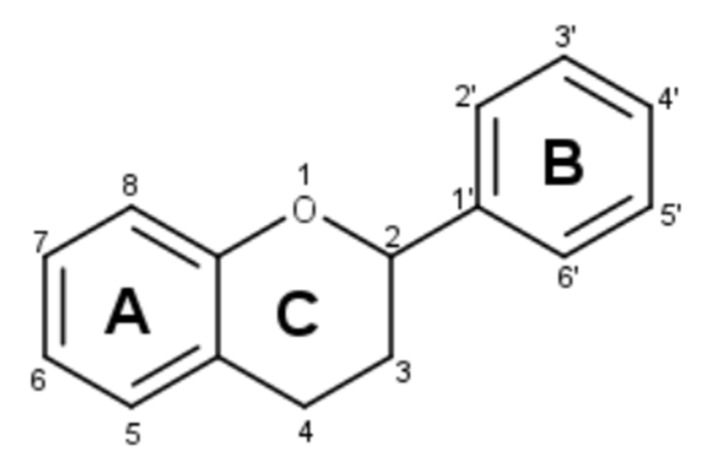
Flavonoid structure.

**Figure 2 molecules-27-03089-f002:**
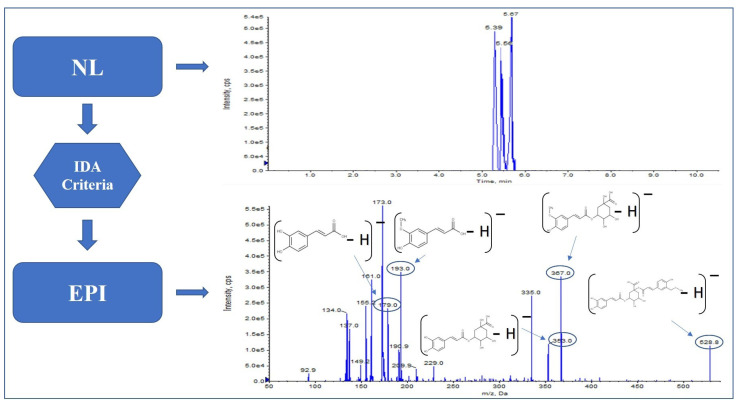
Feruloyl-5-caffeoylquinic acid fragmentation pattern in green coffee.

**Figure 3 molecules-27-03089-f003:**
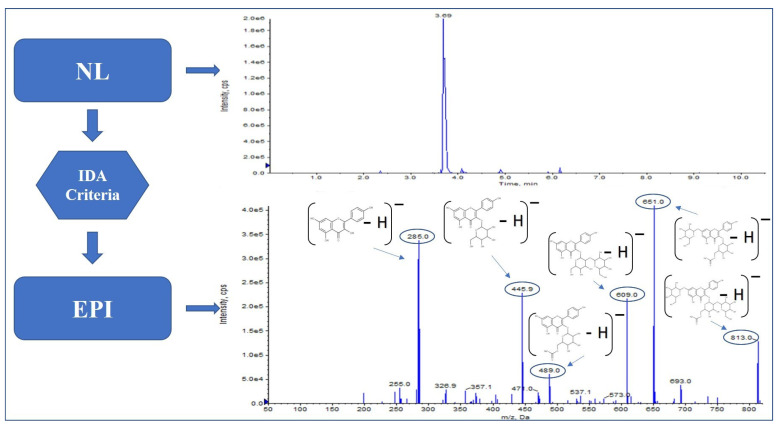
Kaempferol-glucosyl-(1->2)-(6″-acetylgalactoside)-hexoside fragmentation pattern in Saffron.

**Figure 4 molecules-27-03089-f004:**
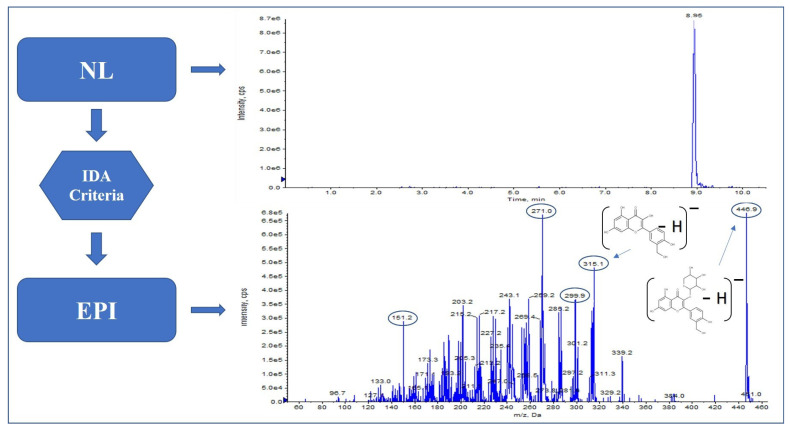
Isorhamnetin-xyloside pattern in hop.

**Table 1 molecules-27-03089-t001:** PCs identified and quantified with the targeted method in green coffee, saffron and hop samples. Data are reported in µg/g, and each matrix was analyzed in three biological replicates, which were extracted in three replicates.

Sample Name	Gallic Acid	Chlorogenic Acid	Epicatechin	Catechin	Caffeic Acid	Vanillic Acid	EGCG	Siringic Acid	Protocatechuic Acid	*p*-Coumaric Acid
Green coffee	41.53 ± 2.5	16,990.00 ± 1,868	20.20 ± 1.8	13.06 ± 1.2	702.60 ± 49.2	1.77 ± 0.7	4.60 ± 1.2	15.21 ± 0.9	51.25 ± 2.0	22.52 ± 7.6
Saffron	11.93 ± 1.4	73.80 ± 5.2	15.67 ± 1.3	<LOQ	24.40 ± 3.2	<LOQ	5.49 ± 1.7	27.76 ± 1.7	<LOQ	15.23 ± 1.6
Hop	<LOQ	912.60 ± 82.2	805.90 ± 84.6	4692.00 ± 516.1	46.35 ± 4.2	105.50 ± 8.4	6.23 ± 0.2	684.00 ± 75.3	840.00 ± 50.4	992.00 ± 89.3
**Sample Name**	**Ferulic Acid**	**Rosmarinic Acid**	**Quercetin**	**Rutin**	**Ellagic Acid**	**Sinapic Acid**	**Luteolin**	**Quercetin-hexoside**	**Isoxanthohumol**	**Xhanthohumol**
Green coffee	275.80 ± 69.3	10.64 ± 2.6	47.89 ± 1.9	77.19 ± 3.9	4.33 ± 0.9	6.71 ± 0.4	<LOQ	4.19 ± 0.9	<LOQ	<LOQ
Saffron	0.60 ± 0.1	3.04 ± 0.1	52.48 ± 3.7	0.90 ± 0.1	4.19 ± 0.7	4.56 ± 0.9	3.84 ± 0.1	7.46 ± 1.1	<LOQ	<LOQ
Hop	168.00 ± 12.4	<LOQ	549.00 ± 89.4	1146 ± 676.2	14.38 ± 1.2	307.00 ± 27.6	831.00 ± 94.8	745.90 ± 82.3	468.00 ± 38.8	1787.00 ± 496.6

**Table 2 molecules-27-03089-t002:** Green coffee, hop and saffron samples analyzed using the proposed NL-IDA-EPI acquisition method. Different isomeric forms were tagged with (I), (II), (III) for caffeoylquinic acid, feruloyl-5-caffeoylquinic acid and diferuloylquinic acid.

	T_r_	M/Z	NL	Main Fragments				Compound Identification
Green coffee	3.34	352.9	[M − H − 162]^−^ or [M − H − 174]^−^	135	161	179	191	Caffeoylquinic acid (I)
	3.76	352.9	[M − H − 162]^−^ or [M − H − 174]^−^	135	161	179	191	Caffeoylquinic acid (II)
	3.77	451	[M – H − 162]^−^	109	125	245	289	Catechin-hexoside
	3.88	352.9	[M − H − 162]^−^ or [M − H − 174]^−^	135	161	179	191	Caffeoylquinic acid (III)
	4.37	366.9	[M − H − 174]^−^ or [M − H − 176]^−^	134	135	173	193	Feruloylquinic acid
	4.58	337	[M − H − 146]^−^ or [M − H − 174]^−^	117	119	163	191	*p*-Coumaroylquinic acid
	4.9	514.9	[M − H − 162]^−^ or [M − H − 174]^−^	161	179	191	353	Dicaffeoylquinic acid (I)
	5.13	514.9	[M − H − 162]^−^ or [M − H − 174]^−^	161	179	191	353	Dicaffeoylquinic acid (II)
	5.39	529	[M − H − 162]^−^ or [M − H − 174]^−^ or [M – H − 176]^−^	179	193	353	367	Feruloyl-5-caffeoylquinic acid (I)
	5.56	529	[M − H − 162]^−^ or [M − H − 174]^−^ or [M – H − 176]^−^	179	193	353	367	Feruloyl-5-caffeoylquinic acid (II)
	5.67	529	[M − H − 162]^−^ or [M − H − 174]^−^ or [M – H − 176]^−^	179	193	353	367	Feruloyl-5-caffeoylquinic acid (III)
	5.92	542.9	[M − H − 174]^−^ or [M − H − 176]^−^	134	173	193	367	Diferuloylquinic acid (I)
	6.13	542.9	[M − H − 174]^−^ or [M − H − 176]^−^	134	173	193	367	Diferuloylquinic acid (II)
	6.3	542.9	[M − H − 174]^−^ or [M − H − 176]^−^	134	173	193	367	Diferuloylquinic acid (III)
Hop	3.34	352.9	[M − H − 162]^−^ or [M − H − 174]^−^	135	161	179	191	Caffeoylquinic acid (I)
	3.76	352.9	[M − H − 162]^−^ or [M − H − 174]^−^	135	161	179	191	Caffeoylquinic acid (II)
	3.88	352.9	[M − H − 162]^−^ or [M − H − 174]^−^	135	161	179	191	Caffeoylquinic acid (III)
	4.6	609	[M − H − 162]^−^ or [M − H − 308]^−^	151	179	271	301	Quercetin-rutinoside
	4.77	592.8	[M − H − 162]^−^ or [M − H − 308]^−^	133	159	229	285	Kaempferol-rutinoside
	5.31	463	[M − H − 174]^−^	151	179	271	301	Quercetin-hexoside
	7.16	396.8	[M − H − 174]^−^	173	207	281	353	Sinapoylquinic acid (I)
	7.44	396.8	[M − H − 174]^−^	173	207	281	353	Sinapoylquinic acid (II)
	7.65	396.8	[M − H − 174]^−^	173	207	281	353	Sinapoylquinic acid (III)
	8.49	433,2	[M − H − 132]^−^	151	179	271	301	Quercetin-xyloside
	8.95	446.9	[M − H − 132]^−^	151	271	300	315	Isorhamnetin-xyloside
	9.84	311	[M − H − 162]^−^	103	135	179	249	Caffeoyltartaric acid
Saffron	3.27	787	[M − H − 146]^−^ or [M − H − 162]^−^ or [M − H − 308]^−^	317	463	479	625	Myricetin-rutinoside-hexoside
	3.38	771	[M − H − 162]^−^	255	285	446	609	Kaempferol-sophoroside-hexoside
	3.47	755	[M − H − 146]^−^ or [M − H − 162]^−^ or [M − H − 308]^−^	255	285	446	593	Kaempferol-rutinoside-hexoside
	3.69	813	[M − H − 162]^−^ or [M − H − 308]^−^	446	489	609	651	Kaempferol-glucosyl-(6″-acetylgalactoside)-hexoside
	3.77	314.6	[M − H − 162]^−^	53	109	135	153	Protocatechuic acid-hexoside
	3.77	609	[M − H − 162]^−^	159	255	285	446	Kaempferol-sophoroside
	4.12	651	[M − H − 146]^−^ or [M − H − 162]^−^ or [M − H − 308]^−^	151	179	447	489	Quercetin-*O*-(6″-acetyl-galactoside)-*O*-rhamnoside
	4.21	448.5	[M − H − 162]^−^	213	231	259	287	Cyanidin-hexoside
	4.21	624.8	[M − H − 162]^−^	151	179	301	463	Quercetin-sophoroside
	4.29	639	[M − H − 162]^−^	151	271	315	477	Isorhamnetin-sophoroside
	4.41	609	[M − H − 146]^−^ or [M − H − 162]^−^ or [M − H − 308]^−^	151	179	271	301	Quercetin-rutinoside
	4.64	623	[M − H − 146]^−^ or [M − H − 162]^−^ or [M − H − 308]^−^	151	271	300	315	Isorhamnetin-rutinoside
	4.89	651	[M − H − 162]^−^	255	285	446	489	Kaempferol-(6″-acetyl-glucoside)-glucoside
	5.02	447	[M − H − 162]^−^	133	159	229	285	Kaempferol-hexoside
	5.1	478	[M − H − 162]^−^	257	274	302	316	Petunidin-hexoside
	5.1	785	[M − H − 146]^−^ or [M − H − 162]^−^ or [M − H − 308]^−^	300	315	477	639	Isorhamnetin-sophoroside-rhamnoside
	5.31	463	[M − H − 162]^−^	151	179	271	301	Quercetin-hexoside
	5.33	432.7	[M − H − 162]^−^	141	188	225	270	Pelargonidin-hexoside
	5,81	562	[M − H − 132]^−^ or [M − H − 162]^−^	117	151	269	431	Apigenin-apiosyl-hexoside
	5.95	593	[M − H − 146]^−^ or [M − H − 162]^−^ or [M − H − 308]^−^	133	159	229	285	Kaempferol-rutinoside
	6.52	329	[M − H − 162]^−^	151	179	271	301	Dimethylquercetin
	7.58	430.8	[M − H − 146]^−^	133	159	229	285	Kaempferol-rhamnoside
	9.84	311	[M − H − 162]^−^	103	135	179	249	Caffeoyl tartaric acid

## Data Availability

Data are available from authors upon reasonable request.

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
