# Peer review of "Predictive Multi Experiment Approach for the Determination of Conjugated Phenolic Compounds in Vegetal Matrices by Means of LC-MS/MS"

_molecules, 2022, doi:10.3390/molecules27103089_

Round 1

Reviewer 1 Report

The authors in this paper presented targeted and partially untargeted approaches to identify and quantify different conjugated forms of polyphenols in different food matrices such as green coffee, Crocus sativus L (saffron) and Humulus lupulus L. (hops). In this study, a novel approach i.e. neutral loss (NL) IDA-Enhanced product ion, implemented on a LIT-QqQmethod was used to identify various flavonoid glycoside derivatives and quinic acid esters with hydroxycinnamic acids in selected food matrices.

The manuscript is generally well written. However, the following issues have to be addressed before this manuscript is suitable for publication.

  1. Line: 51-53 Please use bibliographic references that can support this statement.
  2. Line: 61-64 Please this statement needs a bibliographic reference. Please provide examples of studies in which the listed analytical methods were used to determine polyphenols.
  3. The Materials and Methods section should come before the Results section.
  4. All figures (chromatograms) must be done with better resolution.
  5. There is no information about the samples: origin, quantity, etc.
  6. The manuscript lacks a description of the validation of the method used and the presentation of the basic parameters involved.
  7. The discussion section lacks comparison with studies by other researchers.
  8. Line: 348-349 This sentence is unnecessary.

I appreciate the interest of the authors in the development of this manuscript. It is an interesting topic. The work should be redrafted and supplemented with important aspects, therefore I suggest a major revision.

Author Response

Response to Reviewer comments

Open Review 1

English language and style

( ) Extensive editing of English language and style required
( ) Moderate English changes required
( ) English language and style are fine/minor spell check required
(x) I don't feel qualified to judge about the English language and style

Yes

Can be improved

Must be improved

Not applicable

Does the introduction provide sufficient background and include all relevant references?

( )

(x)

( )

( )

Is the research design appropriate?

( )

(x)

( )

( )

Are the methods adequately described?

( )

( )

(x)

( )

Are the results clearly presented?

( )

( )

(x)

( )

Are the conclusions supported by the results?

( )

(x)

( )

( )

Comments and Suggestions for Authors

The authors in this paper presented targeted and partially untargeted approaches to identify and quantify different conjugated forms of polyphenols in different food matrices such as green coffee, Crocus sativus L (saffron) and Humulus lupulus L. (hops). In this study, a novel approach i.e. neutral loss (NL) IDA-Enhanced product ion, implemented on a LIT-QqQmethod was used to identify various flavonoid glycoside derivatives and quinic acid esters with hydroxycinnamic acids in selected food matrices.

The manuscript is generally well written. However, the following issues have to be addressed before this manuscript is suitable for publication.

  1. Line: 51-53 Please use bibliographic references that can support this statement.

According to the referee suggestion, the following reference was added:

“The most common sugar moieties of flavonoids present in plants are glucose, rhamnose, galactose, apiose, rutinoside, arabinose and glucuronic acid; in fact, flavonoids are usually available in the form of mono, di- and triglycosides [8]”

  1. Xiao, J.; Muzashvili, T.S.; Georgiev, M.I. Advances in the Biotechnological Glycosylation of Valuable Flavonoids. Biotechnol. Adv. 2014, 32, 1145–1156, doi:10.1016/j.biotechadv.2014.04.006.
  1. Line: 61-64 Please this statement needs a bibliographic reference. Please provide examples of studies in which the listed analytical methods were used to determine polyphenols.

According to the referee suggestion, the following references were added:

Note the importance of PCs, there are numerous analytical methods for their determination in food matrices; high performance liquid chromatography (HPLC) coupled with mass spectrometry (MS) is certainly the most used method for their identification and quantification. MS was applied both in low resolution (LRMS) [11,12], using as detector triple quadrupole (QqQ), linear ion trap (LIT) and hybrid quadrupole-linear ion trap mass spectrometers (LIT-QqQ) [13], or in high resolution (HRMS) [14,15], coupled with instruments such as Q-TOF [16,17], Q-Orbitrap and LTQ-Orbitrap [18,19], which is usually used for comprehensive characterization by means of untargeted analysis.

  1. Motilva, M.J.; Serra, A.; Macià, A. Analysis of Food Polyphenols by Ultra High-Performance Liquid Chromatography Coupled to Mass Spectrometry: An Overview. J. Chromatogr. A 2013, 1292, 66–82, doi:10.1016/j.chroma.2013.01.012.
  2. Tohma, H.; Köksal, E.; Kılıç, Ö.; Alan, Y.; Abdullah Yılmaz, M.; Gülçin, Ä°.; Bursal, E.; Alwasel, S.H. RP-HPLC/MS/MS Analysis of the Phenolic Compounds, Antioxidant and Antimicrobial Activities of Salvia L. Species. Antioxidants 2016, 5, 1–15, doi:10.3390/antiox5040038.
  3. López-fernández, O.; Domínguez, R.; Pateiro, M.; Munekata, P.E.S.; Rocchetti, G.; Lorenzo, J.M. Determination of Polyphenols Using Liquid Chromatography–Tandem Mass Spectrometry Technique (LC–MS/MS): A Review. Antioxidants 2020, 9, 1–27, doi:10.3390/antiox9060479.
  4. Santarsiero, A.; Onzo, A.; Pascale, R.; Acquavia, M.A.; Coviello, M.; Convertini, P.; Todisco, S.; Marsico, M.; Pifano, C.; Iannece, P.; et al. Pistacia Lentiscus Hydrosol: Untargeted Metabolomic Analysis and Anti-Inflammatory Activity Mediated by NF- κ B and the Citrate Pathway. Oxid. Med. Cell. Longev. 2020, 2020, doi:10.1155/2020/4264815.
  5. Ballesteros-Vivas, D.; Álvarez-Rivera, G.; Ibáñez, E.; Parada-Alfonso, F.; Cifuentes, A. A Multi-Analytical Platform Based on Pressurized-Liquid Extraction, in Vitro Assays and Liquid Chromatography/Gas Chromatography Coupled to High Resolution Mass Spectrometry for Food by-Products Valorisation. Part 2: Characterization of Bioactive Compound. J. Chromatogr. A 2019, 1584, 144–154, doi:10.1016/j.chroma.2018.11.054.
  6. AydoÄŸan, C. Recent Advances and Applications in LC-HRMS for Food and Plant Natural Products: A Critical Review. Anal. Bioanal. Chem. 2020, 412, 1973–1991.

  1. The Materials and Methods section should come before the Results section.

We followed the journal template, in which Material and Methods section is placed after Results and Discussion.

  1. All figures (chromatograms) must be done with better resolution.

We have uploaded the figures at 500 dpi: probably the low resolution in the pdf file is due to conversion process. Anyway the original files were sent to the editorial office.

  1. There is no information about the samples: origin, quantity, etc.

As suggested by the reviewer, we added the following sentence in the manuscript:

“The matrices used for the analysis of PCs were purchased from local resellers: green coffee and saffron in the form of powder, while hops as inflorescence”

  1. The manuscript lacks a description of the validation of the method used and the presentation of the basic parameters involved.
    Thanks to the reviewer for the observation. The requested information were added:

“The method showed a sensitive and robust quantitative analysis on the target analytes, providing limited of quantifications (LOQs) ranging between (0.0004 and 0.06 ng mg-1). Furthermore, the precision and accuracy of the method was suitable, with values included between ±10% near LOQs.

The validation of the method was performed following the FDA guidelines [36], considering the LOQs, LODs, accuracy, precision and linearity parameters.”

[36] Food and Drug Administration (FDA) Methods, Method Verification and Validation. ORA Lab. Proced. 2014, 1.7.

  1. The discussion section lacks comparison with studies by other researchers.

As suggested, we improved the Discussion section as follows:

“The targeted approach was used to identify and quantify different PCs in the food matrices under examination; this type of analysis is certainly a widely used technique for the characterization of numerous matrices. As reported in the literature, numerous studies have been carried out on green coffee beans [50,51] in which particular attention is paid to phenol acids and their derivatives such as the class of chlorogenic acids [31]; in saffron, where the content of flavonoids and in particular of the conjugated forms has been mainly studied [52,53] and in hop, where xanthones, in particular xanthohumol and isoxanthohumol, are the characteristic analytes [41,54]. With the proposed approach, however, it was possible to characterize numerous compounds belonging to different classes of PCs in a single run; thus allowing to obtain a characterization of the matrices of interest, to which a semi-untargeted analysis has been added to obtain a more comprehensive analysis.

The potential of LIT was exploited for the semi-untargeted analysis of PCs; how-ever, as far as we know, this approach was not used specifically for the 3 proposed matrices and the polyphenols analyzes were mainly conducted with EMS-IDA-EPI and MRM-IDA-EPI acquisition modalities [20-25] and in a targeted list was used for com-pound identification; with the neutral loss, on the other hand, it is possible to identify numerous compounds characterized by the same moiety, allowing to identify with a single run analytes belonging to classes belonging to different classes, as in the case of the neutral fragment 162 Da which allows to detect both compounds that have a glucose moiety in structure, such as the glycosidic derivatives of flavonoids, and the derivatives of caffeic acid.”

“3.2 Polyphenols identification

The two proposed approaches were tested on food matrices to evaluate their effectiveness. In the green coffee sample it was possible to identify mainly derivatives of hydroxycinnamic acids were identified as the isomers of caffeoylquinic acid (chloro-genic acid, cryptochlorogenic acid and neochlorogenic acid) and more complex forms such as 1,5-dicaffeoylquinic (cynarin) acid and 3,4 -dicaffeoylquinic acid (isochloro-genic acid) as well as derivatives of feruloylquinic acid, as reported in the literature [31,55,56]. In the saffron sample, an increase in conjugated forms of flavonoids such as kaempferol-hexoside, pelargonidin-hexoside, petunidin-hexoside and more complex forms such as isorhamnetin-sophoroside, known in the literature, was noted [57–59]. Finally, the MRM analyzes carried out on the hop sample allowed to identify and quantify xanthohumol and isoxanthohumol [41], characteristic of this matrix, while the semi-untargeted showed the presence of phenolic acids such as the three isomers of caffeoylquinic acid [60] and flavonoid derivatives. such as kaempferol-rutinoside [61].

It can therefore be said that the proposed methods are able to provide reliable results comparable with the information present in the literature, thus allowing to carry out characterizations of the matrices of interest with and without the use of standards.”

  1. Line: 348-349 This sentence is unnecessary.

Thanks to the reviewer for this comment. It was a mistyping, so the sentence was removed

I appreciate the interest of the authors in the development of this manuscript. It is an interesting topic. The work should be redrafted and supplemented with important aspects, therefore I suggest a major revision.

Reviewer 2 Report

The originality of this manuscript is low, the results are not applicable and the whole work seems to be a low-quality copy of their previous article (https://doi.org/10.1016/j.chroma.2021.462315). It cannot be published in "Molecules." The authors claim to have developed semi-untargeted method using LIT-QqQ LC-MS for identification and quantification of polyphenols in food matrices: green coffee, saffron, hop. The authors' claims are very much over the top. Not only do they not contribute anything new, but the quality of their work seems to be at a very low level in relation to the existing literature.

Chlorogenic acids are a family of esters formed between certain cinnamic acids and quinic acid (Clifford et al., 2003), and are the main polyphenolic constituents of the coffee beans. However, the authors did not cite the scientific papers of Michael Clifford, who dedicated his scientific carrier on detailed MS/MS characterization of chlorogenic acids in coffee beans and other plant objects. One of his well cited papers is “Hierarchical Scheme for LC-MSn Identification of Chlorogenic Acids”. In the conclusion section the authors indicated, in their opinion, newly detected compounds (line 376-379) - isomers of caffeoylquinic acid…..in green coffee, but these compounds were already characterized by Clifford almost 20 years ago.

The neutral losses and MS fragment patterns of hydroxycinnamic acids and flavonoids conjugates are well known and widely published. It was actual scientific topics in late 90s and yearly 2000-2010. However, authors did mistakes in the identification of polyphenols and interpretation of their results: for the identification of phenolics, the authors used the NL (neutral loss)-IDA_EPI mode. NL were represented by (lines 333-335) 162 amu a hexose and quinoyl groups, 191 amu for caffeoyl groups, etc. It should be 162 amu for hexose and caffeoyl unit, and 174 for quinoyl unit. However, 191 is not a neutral loss, it’s m/z of quinic acid [M-H]-. The presented data in the Table 2 did not fit with the NLs and main fragments of the identified compounds with the literature data Clifford et al., Kolodziejczyk-Czepas et al., 2021 10.1016/j.phytochem.2021.112861. Herein, p-Coumaroylquinic acid doesn’t release MS fragment 161, and m/z 179 is a product ion of caffeoylquinic acid instead of feruloylquinic acid as authors described.

Another aspect of this paper are performed quantitative analyses. The authors cleverly concealed information about the method of quantification (no calibration curves, limits of detection/quantitation, linearity, etc.): line 114: “the identification and quantification of PCs in the three plant matrices, using standards with the previously described method” without citing it. Furthermore, the analytical standards used (line 288-291) in section “Materials and Methods” did not correspond to the detected analytes (lines 276-280) and quantified (table 1) in this manuscript.

The Figures 1 and 2 did not correspond with their titles, and Fig. 4 is not even present in the manuscript.

Following, you can find more detailed comments:

Line 51-55: The authors did not mention about C-glycosides. C-glycosides are common flavonoids.

Line 101-104: the chemical group of resins, should be more precise as prenylated phloroglucinol derivatives (bitter acids).

Lines 93-96: The reference [23] is not enough for supporting characterization of chemical composition of coffee beans. Sanlier et al., 2019 gave the main accent on the nutritional, health and toxicological effects of coffee.  The authors did not cite the scientific papers of Michael Clifford, who dedicated his scientific carrier on LC-MS identification of polyphenols i.e. chlorogenic acids in coffee beans and other plant objects.

 Line 96: I do not agree with term chlorogenic and caffeic acid. Citing Clifford M.N. et al., 2003: “chlorogenic acids (CGA) are a family of esters formed between certain cinnamic acids and quinic acid.”

Line 100: The reference [24] did not illustrate the chemical composition of Saffron. Herein the detailed information on phytochemical composition of Saffron https://doi.org/10.1016/j.phytochem.2019.02.004

In the introduction section, authors cited the references that related more with biological aspects of the plant objects which is not consistent with their purpose and title.

Line 120: the coffee beans contain several isomers of caffeic and ferulic acids. It’s not a single compound.

Line 288: In the section of “Materials and Methods 4.1. Chemicals”: 1) the chemical name of chlorogenic acid standard should be given. 2) the list of the standards (14) did not appropriate with the standards that were determined in the vegetal matrices (20) – Table 1. (the section “Results”).

Line 330: there is no Figure 4, that was cited in the text.

Line 348-349. This statement about the conclusion section is unnecessary.

Author Response

Response to Reviewer comments

open Review 2

English language and style

( ) Extensive editing of English language and style required
( ) Moderate English changes required
( ) English language and style are fine/minor spell check required
(x) I don't feel qualified to judge about the English language and style

Yes

Can be improved

Must be improved

Not applicable

Does the introduction provide sufficient background and include all relevant references?

( )

( )

( )

(x)

Is the research design appropriate?

( )

( )

( )

(x)

Are the methods adequately described?

( )

( )

( )

(x)

Are the results clearly presented?

( )

( )

( )

(x)

Are the conclusions supported by the results?

( )

( )

( )

(x)

Comments and Suggestions for Authors

The originality of this manuscript is low, the results are not applicable and the whole work seems to be a low-quality copy of their previous article (https://doi.org/10.1016/j.chroma.2021.462315). It cannot be published in "Molecules." The authors claim to have developed semi-untargeted method using LIT-QqQ LC-MS for identification and quantification of polyphenols in food matrices: green coffee, saffron, hop. The authors' claims are very much over the top. Not only do they not contribute anything new, but the quality of their work seems to be at a very low level in relation to the existing literature.

Chlorogenic acids are a family of esters formed between certain cinnamic acids and quinic acid (Clifford et al., 2003), and are the main polyphenolic constituents of the coffee beans. However, the authors did not cite the scientific papers of Michael Clifford, who dedicated his scientific carrier on detailed MS/MS characterization of chlorogenic acids in coffee beans and other plant objects. One of his well cited papers is “Hierarchical Scheme for LC-MSn Identification of Chlorogenic Acids”. In the conclusion section the authors indicated, in their opinion, newly detected compounds (line 376-379) - isomers of caffeoylquinic acid..in green coffee, but these compounds were already characterized by Clifford almost 20 years ago.

We apologize for some unclear sentences in the manuscript especially about the aim of the work. Our intent was to develop and validate a semi-untargeted method (neutral loss IDA-Enhanced product ion approach) able to identify polyphenols compounds in different complex matrix. In the conclusion section (line 376-379) we wanted to highlight the results about the detection of compounds not included in MRM list, not the identification of new compounds.

For sake of clarity, the sentence was modified as follows:

“In particular, several compounds, not belonging to the target list, were detected in the selected matrices: isomers of caffeoylquinic acid (chlorogenic acid, cryptochlorogenic acid and neochlorogenic acid) and the isomers of feruloylquinic acid have been identified in green coffee, in addition to their complex forms (dicaffeoylquinic acid and diferuloylquinic acid); conjugates of flavonoids such as apigenin-apiosyl-hexoside and kaempferol-glucosyl-(1->2)-(6''-acetylgalactoside)-hexoside have been identified in saffron; while both conjugated forms of the quinic acid esters such as the isomers of sinapoylquinic acid and glycosylated flavonoids like isorhamnetin-xyloside have been identified in hop.”

Moreover in discussion section we added the following paragraph to compare our results with the literature in order to highlight the reliability of the method:

“3.2 Polyphenols identification

The two proposed approaches were tested on food matrices to evaluate their effectiveness. In the green coffee sample it was possible to identify mainly derivatives of hydroxycinnamic acids were identified as the isomers of caffeoylquinic acid (chlorogenic acid, cryptochlorogenic acid and neochlorogenic acid) and more complex forms such as 1,5-dicaffeoylquinic (cynarin) acid and 3,4 -dicaffeoylquinic acid (isochlorogenic acid) as well as derivatives of feruloylquinic acid, as reported in the literature [31,55,56]. In the saffron sample, an increase in conjugated forms of flavonoids such as kaempferol-hexoside, pelargonidin-hexoside, petunidin-hexoside and more complex forms such as isorhamnetin-sophoroside, known in the literature, was noted [57–59]. Finally, the MRM analyzes carried out on the hop sample allowed to identify and quantify xanthohumol and isoxanthohumol [41], characteristic of this matrix, while the semi-untargeted showed the presence of phenolic acids such as the three isomers of caffeoylquinic acid [60] and flavonoid derivatives, such as kaempferol-rutinoside [61].

It can therefore be said that the proposed methods are able to provide reliable results comparable with the information present in the literature, thus allowing to carry out characterizations of the matrices of interest with and without the use of standards.”

The neutral losses and MS fragment patterns of hydroxycinnamic acids and flavonoids conjugates are well known and widely published. It was actual scientific topics in late 90s and yearly 2000-2010. However, authors did mistakes in the identification of polyphenols and interpretation of their results: for the identification of phenolics, the authors used the NL (neutral loss)-IDA_EPI mode. NL were represented by (lines 333-335) 162 amu a hexose and quinoyl groups, 191 amu for caffeoyl groups, etc. It should be 162 amu for hexose and caffeoyl unit, and 174 for quinoyl unit. However, 191 is not a neutral loss, it’s m/z of quinic acid [M-H]-. The presented data in the Table 2 did not fit with the NLs and main fragments of the identified compounds with the literature data Clifford et al., Kolodziejczyk-Czepas et al., 2021 10.1016/j.phytochem.2021.112861. Herein, p-Coumaroylquinic acid doesn’t release MS fragment 161, and m/z 179 is a product ion of caffeoylquinic acid instead of feruloylquinic acid as authors described.

We thanks the reviewer for this comment: it was a mistake due to a carelessness during the editing of paper. We have corrected it in the manuscript and in the table.

“For the identification of the conjugated forms of the PCs, NL-IDA-EPI scans were then used with common losses both for the glycosidic forms of flavonoids and quinic acid esters of hydroxycinnamic acid, with neutral loss m/z 132, 146, 162, 174, 176 and 308, corresponding to the mass of a pentose unit, rhamnose and coumaroyl units, a hexose and caffeoyl unit, quinoyl units, feruloyl unit, rutinose units, respectively; re-sults are shown in Table 2. in according to Clifford et al. [31] and Ko-lodziejczyk-Czepas et al. [42].”

“Indeed, as reported in Table 2. the NL-IDA-EPI strategy detected several neutral unit losses of 146 amu corresponding to [M−H−coumaroyl]- unit, 162 amu corre-sponding to [M−H−caffeoyl]- unit, 174 amu responding to [M−H−quinoyl]- unit, 176 amu responding to [M−H−feruloyl]- unit.”

“In the NL survey scan, several common losses were selected following neutral losses of quinic acid esters of hydroxycinnamic acids and glycosidic derivatives of flavonoids. For the quinic acid esters, the following values were used: 146 amu for [M−H−coumaroyl]- unit, 162 amu for [M−H−caffeoyl]- unit, 174 amu for [M−H−quinoyl]- unit, 176 amu for [M−H−feruloyl]- unit. For the identification of the glycosidic derivatives the following were used: the 132 amu for the [M−H−pentose]- unit, the 146 amu also for [M−H−rham]- unit, 162 amu also for hexose [M−H−hexose]- unit and 308 amu for [M−H−rut]- unit.”

Also, the table was modified as suggested.

Another aspect of this paper are performed quantitative analyses. The authors cleverly concealed information about the method of quantification (no calibration curves, limits of detection/quantitation, linearity, etc.): line 114: “the identification and quantification of PCs in the three plant matrices, using standards with the previously described method” without citing it.

The information about quantification was reported in our previous work. For this reason, we added the reference in the following sentence:

“MRM analysis was used for the identification and quantification of PCs in the three plant matrices, using standards with the previously described method [35]; results are shown in Table 1.”

[35] Oliva, E.; Viteritti, E.; Fanti, F.; Eugelio, F.; Pepe, A.; Palmieri, S.; Sergi, M.; Compagnone, D. Targeted and Semi-Untargeted Determination of Phenolic Compounds in Plant Matrices by High Performance Liquid Chromatography-Tandem Mass Spectrometry. J. Chromatogr. A 2021, 1651, 462315, doi:10.1016/j.chroma.2021.462315.

Furthermore, the information about method validation were added:

“The method showed a sensitive and robust quantitative analysis on the target analytes, providing limited of quantifications (LOQs) ranging between (0.0004 and 0.06 ng mg-1). Furthermore, the precision and accuracy of the method was suitable, with values included between ±10% near LOQs.

The validation of the method was performed following the FDA guidelines [36], considering the LOQs, LODs, accuracy, precision and linearity parameters.”

[36] Food and Drug Administration (FDA) Methods, Method Verification and Validation. ORA Lab. Proced. 2014, 1.7.

Furthermore, the analytical standards used (line 288-291) in section “Materials and Methods” did not correspond to the detected analytes (lines 276-280) and quantified (table 1) in this manuscript.

Thanks for the comment, we apologize for the missing information about the standards. For this reason, we have modified the text as follows:

“The standards used of PCs used in our research were: gallic acid, OH-tyrosol, protocatechuic acid, (-)-epigallocatechin (EGC), 3-OH-benzoic acid, tyrosol, chlorogenic acid (3-O-caffeoylquinic acid), epicatechin, caffeic acid, vanillic acid, catechin, (-)-epigallocatechin gallate (EGCG), siringic acid, orientin (luteolin-8-glucoside), rutin, p-coumaric acid, hyperoside (quercetin-3-d-galactoside), isoquercetin (querce-tin-3-b-d-glucoside), ferulic acid, hesperidin, rosmarinic acid, oleuropein, o-coumaric acid, sinapic acid, myricetin, luteolin, quercetin, trans-cinnamic acid, naringenin, isoxhanthohumol, apigenin, diosmetin (luteolin-4-methyl ether), kaempferol, xhanthohumol”

The Figures 1 and 2 did not correspond with their titles, and Fig. 4 is not even present in the manuscript.

We apologize for the mistake: Figure 4 was moved to Graphical abstract, but we forgot to update the text. Now we have corrected also the captions in the manuscript and amended the disalignment.

Following, you can find more detailed comments:

Line 51-55: The authors did not mention about C-glycosides. C-glycosides are common flavonoids.

Done.

Line 101-104: the chemical group of resins, should be more precise as prenylated phloroglucinol derivatives (bitter acids).

We have just listed some further components of the selected matrices. Now we have removed this sentence in order to simplify the introduction section.

Lines 93-96: The reference [23] is not enough for supporting characterization of chemical composition of coffee beans. Sanlier et al., 2019 gave the main accent on the nutritional, health and toxicological effects of coffee.  The authors did not cite the scientific papers of Michael Clifford, who dedicated his scientific carrier on LC-MS identification of polyphenols i.e. chlorogenic acids in coffee beans and other plant objects.

As suggested, we modified the manuscript as follows:

“Green coffee is a form of fruit of raw coffee, unroasted, unprocessed and natural, it is mainly used for weight loss thanks to its high antioxidant properties and other health properties due to the abundance of phytochemicals such as caffeine and some polyphenols, such as chlorogenic and caffeic acids (CGA) [30,31]”

  1. Clifford, M.N.; Johnston, K.L.; Knight, S.; Kuhnert, N. Hierarchical Scheme for LC-MSn Identification of Chlorogenic Acids. J. Agric. Food Chem. 2003, 51, 2900–2911, doi:10.1021/jf026187q.

 Line 96: I do not agree with term chlorogenic and caffeic acid. Citing Clifford M.N. et al., 2003: “chlorogenic acids (CGA) are a family of esters formed between certain cinnamic acids and quinic acid.”

Thanks for the comment: in accordance with the referee suggestion we modified the manuscript, as follows:

“Green coffee is a form of fruit of raw coffee, unroasted, unprocessed, and natural, known mainly for its high antioxidant properties, due to the presence of numerous phenolic compounds, in particular phenolic acids and their derivatives, such as chlorogenic acids (CGA) [30,31].”

Line 100: The reference [24] did not illustrate the chemical composition of Saffron. Herein the detailed information on phytochemical composition of Saffron https://doi.org/10.1016/j.phytochem.2019.02.004

According to reviewer suggestion we added the reference in the following sentence:

“Saffron, produced by the dried stigmas of Crocus sativus L., a member of the Iridaceae family, is mainly composed of carbohydrates including starch, which reducing sugars, gums, pectin, pentosans, dextrins and polyphenols [32]; it is an agricultural product of great value that is used exclusively in the kitchen to give flavour, colour and aroma to food [33].”

In the introduction section, authors cited the references that related more with biological aspects of the plant objects which is not consistent with their purpose and title.

Thanks to the reviewer for the comment. For sake of clarity we modified the manuscript as follows:

“In this work, targeted and semi-untargeted approaches are presented for identifying and quantifying different conjugated forms of PCs in different food matrices: green coffee, Crocus sativus L (saffron) and Humulus lupulus L. (hop) were chosen to test the proposed strategy. Green coffee is a form of fruit of raw coffee, unroasted, unprocessed, and natural, known mainly for its high antioxidant properties, due to the presence of numerous phenolic compounds, in particular phenolic acids and their derivatives, such as chlorogenic acids (CGA) [30,31]. Even saffron, produced by the dried stigmas of Crocus sativus L., a member of the Iridaceae family, is a matrix rich in polyphenols [32]; in particular, the stigma is rich in numerous derivatives of flavonoids [33].

Line 120: the coffee beans contain several isomers of caffeic and ferulic acids. It’s not a single compound.

Thanks for the comment. The results explained in the paragraph 2.1.1. were referred to the target analysis, so we quantified only the compounds related to analytical standards included in the presented method. For this reason, in these results we are not taking into account other isomers not corresponding to the same retention time of the considered analytical standards.

Line 288: In the section of “Materials and Methods 4.1. Chemicals”: 1) the chemical name of chlorogenic acid standard should be given. 2) the list of the standards (14) did not appropriate with the standards that were determined in the vegetal matrices (20) – Table 1. (the section “Results”).

The chemical name of chlorogenic acid standard has been now reported in the standard list. The standard list has been amended: now it matches with the data in Table 1.

Line 330: there is no Figure 4, that was cited in the text.

Thanks for the comment, we modified the manuscript accordingly.

Line 348-349. This statement about the conclusion section is unnecessary.

Thanks for the comment: it was a mistyping

Reviewer 3 Report

The manuscript focus on the development and application of a mass spectrometric method for the identification of conjugated phenolic compounds in various vegetal matrices (green coffee, saffron and hop). Although, high resolution mass spectrometry techniques as TOF-MS and Orbitrap MS are usually used in predictive approaches, in this paper authors demonstrate the potential of the quadrupole linear ion trap hybrid mass spectrometer (LIT-QqQ) in this application. Information dependent acquisition (IDA) approach was successfully applied for a semi-untargeted analysis and elucidation of fragmentation pattern of the conjugated polyphenols. The topic is actual and significant in respect to current concerns in the development of nutraceutics.  

I recommend the paper for publication after a minor revision (please see the following observation)

Observations:

R 73: I would suggest the replacement of the word ‘’interesting’’ with ‘’structural’’, as the identification of the compounds without reference standards is based on the fragmentation pattern and structural information

R 348: remove the sentence: ‘’This section is not mandatory but can be added to the manuscript if the discussion is 348 unusually long or complex’’

  • Please add the mass error (mass accuracy) or mass tolerance window used for the extraction of the selected ions. It is an important parameter for reliable data.

Author Response

Response to Reviewer comments

Open Review 3

English language and style

( ) Extensive editing of English language and style required
( ) Moderate English changes required
( ) English language and style are fine/minor spell check required
(x) I don't feel qualified to judge about the English language and style

Yes

Can be improved

Must be improved

Not applicable

Does the introduction provide sufficient background and include all relevant references?

( )

(x)

( )

( )

Is the research design appropriate?

(x)

( )

( )

( )

Are the methods adequately described?

(x)

( )

( )

( )

Are the results clearly presented?

(x)

( )

( )

( )

Are the conclusions supported by the results?

(x)

( )

( )

( )

Comments and Suggestions for Authors

The manuscript focus on the development and application of a mass spectrometric method for the identification of conjugated phenolic compounds in various vegetal matrices (green coffee, saffron and hop). Although, high resolution mass spectrometry techniques as TOF-MS and Orbitrap MS are usually used in predictive approaches, in this paper authors demonstrate the potential of the quadrupole linear ion trap hybrid mass spectrometer (LIT-QqQ) in this application. Information dependent acquisition (IDA) approach was successfully applied for a semi-untargeted analysis and elucidation of fragmentation pattern of the conjugated polyphenols. The topic is actual and significant in respect to current concerns in the development of nutraceutics.  

I recommend the paper for publication after a minor revision (please see the following observation)

Observations:

R 73: I would suggest the replacement of the word ‘’interesting’’ with ‘’structural’’, as the identification of the compounds without reference standards is based on the fragmentation pattern and structural information

The sentence was modified following the reviewer suggestion

R 348: remove the sentence: ‘’This section is not mandatory but can be added to the manuscript if the discussion is 348 unusually long or complex’’

Done, it was a mistyping

Please add the mass error (mass accuracy) or mass tolerance window used for the extraction of the selected ions. It is an important parameter for reliable data.

Thanks for the comment. The following sentence was added:

“The instrumental parameters needed a fine tuning in order to obtain the aimed performances: for example, the IDA threshold was set at 5000 cps, above which the fragmentation patterns of each analyte were collected by EPI of the 4 most intense peaks; the LIT fill time has been set to 25 ms and the mass tolerance was set at 0.25 Da.”

Round 2

Reviewer 1 Report

I appreciate the efforts of the authors in making the suggested changes to improve the article. All sections were improved with suggestions. In general, the manuscript is clear and fluent, I do not observe discrepancies. Once the changes are made, I recommend it for publication.

Author Response

Thank you for positive evaluation of our revised manuscript.

Reviewer 2 Report

Generally, the authors responded to my requests.

However, the authors did not indicate the number of biological/technical repetitions, thus there is no possibility to compare the results or asses their validity.

Some comments are also given below:

Line 98. „Even saffron…” – Adverb „Eeven” doesn’t have sense in this sentence.

Line 166: …(Figure 2.). – The point “.” is used twice.

Line 253-261: this statement is so complex and difficult for understanding, i.e.: “belonging to classes belonging to “

p-coumaric, trans-cinnamic should be written in Italic.

Table 1 doesn't include the meaning of standard deviation.

Author Response

Open Review

English language and style

( ) Extensive editing of English language and style required
( ) Moderate English changes required
( ) English language and style are fine/minor spell check required
(x) I don't feel qualified to judge about the English language and style

Yes

Can be improved

Must be improved

Not applicable

Does the introduction provide sufficient background and include all relevant references?

(x)

( )

( )

( )

Are all the cited references relevant to the research?

(x)

( )

( )

( )

Is the research design appropriate?

( )

( )

( )

( )

Are the methods adequately described?

( )

(x)

( )

( )

Are the results clearly presented?

( )

( )

(x)

( )

Are the conclusions supported by the results?

( )

( )

( )

( )

Comments and Suggestions for Authors

Generally, the authors responded to my requests.

However, the authors did not indicate the number of biological/technical repetitions, thus there is no possibility to compare the results or asses their validity.

We modified the manuscript as suggested, by adding the information in the caption of Table 1:

“Table 1. PCs identified and quantified with the targeted method in green coffee, saffron and hop samples. Data are reported as µg/g each matrix was analyzed in three biological replicates which were extracted in three replicates.”

Some comments are also given below:

Line 98. „Even saffron…” – Adverb „Eeven” doesn’t have sense in this sentence.

We modified the manuscript as suggested.

Line 166: …(Figure 2.). – The point “.” is used twice.

This typing is in accordance with guidelines of the journal. So we have not modified the manuscript as suggested.

Line 253-261: this statement is so complex and difficult for understanding, i.e.: “belonging to classes belonging to “

To simplify the sentence for understanding the manuscript was modified as follows:

“The proposed approach allowed, with a single analysis, to identify and quantitate numerous PCs belonging to the different classes; the information deriving from the semi-untargeted analysis allowed to obtain a more complete characterization of the matrices.

The potential of LIT was exploited for the semi-untargeted analysis of PCs; however, as far as we know, this approach was not used for the polyphenols analyses in the 3 proposed matrices; some studies were conducted with EMS-IDA-EPI and MRM-IDA-EPI acquisition modes [20-25] and a targeted list was used for compound identification.

In our work, with the neutral loss scan, it was possible to identify numerous compounds characterized by the same moiety, but belonging to different classes: i.e. the neutral fragment 162 Da allowed the detection of compounds with a glucose moiety in structure, such as the glycosidic derivatives of flavonoids, and the derivatives of caffeic acid.”

p-coumaric, trans-cinnamic should be written in Italic.

Done

Table 1 doesn't include the meaning of standard deviation.

As suggested the standard deviation was added to table 1